# Oral Cardiac Drug–Gut Microbiota Interaction in Chronic Heart Failure Patients: An Emerging Association

**DOI:** 10.3390/ijms25031716

**Published:** 2024-01-31

**Authors:** Ioannis Paraskevaidis, Alexandros Briasoulis, Elias Tsougos

**Affiliations:** 1Division of Cardiology, Hygeia Hospital, Erithrou Stavrou 4, 15123 Athens, Greece; tsougos@yahoo.com; 2Heart Failure Subdivision, Department of Clinical Therapeutics, Alexandra Hospital, Faculty of Medicine, National and Kapodistrian University of Athens, Vassilisis Sofias 80, 11528 Athens, Greece; alexbriasoulis@gmail.com

**Keywords:** gut microbiota, heart failure, drug interaction

## Abstract

Regardless of the currently proposed best medical treatment for heart failure patients, the morbidity and mortality rates remain high. This is due to several reasons, including the interaction between oral cardiac drug administration and gut microbiota. The relation between drugs (especially antibiotics) and gut microbiota is well established, but it is also known that more than 24% of non-antibiotic drugs affect gut microbiota, altering the microbe’s environment and its metabolic products. Heart failure treatment lies mainly in the blockage of neuro-humoral hyper-activation. There is debate as to whether the administration of heart-failure-specific drugs can totally block this hyper-activation, or whether the so-called intestinal dysbiosis that is commonly observed in this group of patients can affect their action. Although there are several reports indicating a strong relation between drug–gut microbiota interplay, little is known about this relation to oral cardiac drugs in chronic heart failure. In this review, we review the contemporary data on a topic that is in its infancy. We aim to produce scientific thoughts and questions and provide reasoning for further clinical investigation.

## 1. Introduction

Heart failure is a severe and harmful syndrome with high mortality and morbidity rates, that are higher among the older population, but not yet well defined since the true prevalence and incidence of this syndrome is not well established. This is because there are varying results depending on geographic location, sex distribution, aging, different phenotype presentation, and stage of severity. To make it more complex and difficult to determine, many studies use different diagnostic criteria, while others use a different study design [1]. Therefore, the diversity and heterogeneity of the incidence and survival is a fact. The aforementioned diversity may be affected by the gut microbiome, which interferes with medical treatment in heart failure patients.

Although a large number of reports have been published showing the interplay of intestinal microbiota with other systems, there are studies that strongly dispute this concept, characterizing this relation as ‘myth and misconception that lacks a solid evidence base’ [2]. Regardless of the different opinions and statements, it remains a fact that exercise training, a Mediterranean diet [3], and lifestyle habits, among other variables, reduce cardiovascular morbidity and mortality rates. Regular exercise training can positively influence the intestinal microbiota bioavailability [3], the human lipid profile, metabolic status, and immune activity, which are all risk factors for cardiovascular diseases [3]. It seems that lifestyle changes can alter the intestinal microbiota environment, leading to either a beneficial [4] or even harmful effect on the cardiovascular system [5,6]. It is of note that, when following a diet rich in fatty acids, gut microbiota in collaboration with hepatic bile enzymes can produce rather harmful effects in the heart metabolite, namely trimethylamine N-oxide (TMAO). Several reports demonstrate the deleterious effect of TMAO production and its relation to major cardiovascular events including atherogenesis, platelet dysfunction, stroke, heart failure, and, most importantly, death [7,8,9]. Accordingly, in patients with chronic heart failure after myocardial infarction, it has been shown that the risk of a major cardiovascular event is increased in the presence of high TMAO levels [10]. On the other hand, the short-chain fatty acids produced by gut microbiota protect mitochondrial DNA, establishing a normal ATP concentration and thus serving the energetic needs of the cardiomyocytes, the pathophysiological base of heart failure syndrome [11]. Low levels are related to better outcomes in HF patients with a reduced ejection [12], and are indices of cardiac fibrosis, hypertrophy, vascular tone [13,14], gut-barrier function, and insulin sensitivity [15,16]. Intestinal gut microbiota and mucosa dysfunction in patients with chronic heart failure vary according to the different phases of the syndrome and are related to the grade of heart failure severity being more compromised in patients with stage C, D or NYHA grade III-IV, than in those with stage A, B or NYHA grade I-II heart failure. It depends on the degree of neuro-humoral and adrenergic stimulation due to low cardiac output, and increased filling pressures [17]. These pathophysiologic reactions lead to gut-wall ischemia, congestion, and chronic low-grade inflammation, thus creating a vicious feedback cycle between the heart and several organs (kidneys, liver). Multiorgan involvement may further impair the already affected gut intestine–microbiota environment, intestinal mucosa permeability, and oral drug–gut microbiota interplay [18].

Intestinal gut microbiota constitute an active ‘organ’ and produce several other metabolites including Lipopolysaccharides (endotoxins), Phenylacetyl glutamine (PAGln), and phenylacetylglycine (PAGly), involved in the gut-barrier function, inflammation, cardiac contractility, insulin resistance, endothelial function, platelet function and thrombosis, decreased cell viability, and myocardial contraction [19,20,21].

Considering the available data, it is clear that intestinal microbiota can positively or negatively affect the health status of a person by changing the human risk factors, and hence predisposing (or not) the individual to cardiovascular diseases and making him/her more or less vulnerable. In this respect, it has been suggested that the metabolites produced by gut microbiota may be altered or can alter drug pharmacokinetics and pharmacodynamics [22,23], Figure 1.

There is a large number of scientific works suggesting a relationship between drugs and gut microbiota [24,25,26], suggesting heterogeneous results (improvement, deterioration, neutral effects). Proton-pump inhibitors, metformin, selective serotonin reuptake inhibitors, laxatives, immune checkpoint inhibition, and antibiotics affect the gut microbiota and metabolites [27]. Interestingly, the relationship between oral medications and the intestinal environment is bidirectional, as more than 24% of non-antibiotic drugs affect the gut microbiota and its metabolic products [28,29].

Although there are several published scientific reports related to the interaction between orally administrated drugs and gut microbiota, little is known about the effects of cardiac drug delivery on the intestinal ‘laboratory’ in patients with cardiovascular diseases, and the possible consequences for therapeutic treatment, although a bidirectional path between drugs and the gut microbiome has been suggested [30]. This raises several unanswered questions. For instance, is the reported resistance to insulin [31,32] or the antiplatelet treatment [33,34] unique to these drug classes, or is it observed in other oral cardiovascular medicines? In the field of heart failure, the residual mortality and morbidity risk, despite the optimization of medical therapy, could be affected by drug–gut microbiota interactions. Improvements in cardiac hemodynamics, as observed in patients with left-ventricular-assist devices and heart transplantations, are associated with a reduction in gut diversity, endotoxemia, inflammation, and oxidative stress [35]. Considering the limited available data, in this manuscript we will try to shed light on the interaction between the gut microbiota and the oral drugs used in cardiovascular medicine, and, more specifically, in patients with chronic heart failure. Moreover, we will focus on oral drug administration and not on injectable drugs, since the former are the main therapeutic approach in everyday clinical practice in patients with chronic heart failure.

## 2. Oral Drugs in Heart Failure

According to the most recent chronic heart failure guidelines, in patients with reduce ejection fraction, the recommended treatment comprises the following oral medicines [36]: angiotensin-converting enzyme inhibitor (ACE-I), or angiotensin-receptor blocker (ARB), angiotensin receptor-neprilysin inhibitor (ARNI), Beta-blocker, mineralocorticoid receptor antagonist (MRA), Dapagliflozin/Empagliflozin (SGLT2), and loop diuretics when indicated. For patients with mildly reduced or preserved ejection fraction, quadruple therapy could be utilized but is not strongly recommended. Despite the advances in medical therapy, there is substantial residual morbidity and mortality risk [36].

### 2.1. Renin–Angiotensin–Aldosterone System Inhibitors

The Renin–Angiotensin–Aldosterone System (RAS) is one of the major pillars for the homeostatic status of the cardiovascular system, and its inhibition is one of the corner stones for the treatment of heart failure, since its inappropriate activation leads to structural remodeling, cardiomyocyte damage and, hence, an impairment of cardiac function. The RAS is a paramount component and is present not only in the kidneys and the lungs but can be found in different organs such as the brain, heart, skeletal muscles, etc., including the digestive organs. These components are in collaboration between them through their autocrine and paracrine action, and interact with endocrine RAS on various levels [37,38]. Concerning the digestive RAS component, it must be noted that it stands by the intestine and is closely related to gut microbiota behavior. The intestinal RAS component is part of the homeostatic process, including glycemic, electrolyte equilibrium and other functions, and can be altered or alter RAS-targeted therapy in patients with chronic heart failure, by modifying the gut–microbiome environment [30,39]. Indeed, it has been shown that experimental animals germ-free infused with angiotensin II have a better behavior on arterial blood pressure and showed less cardiac fibrosis [40]. As has been previously pointed out, the intestinal microbiota are an active endocrine ‘organ’ and produce several bacterial metabolites that affect several organs, through their close collaboration with the RAS components in several organs [36]. Short-chain fatty acids, butyrate acetate or propionate, produced in the intestine, can modulate kidney-local RAS by inhibiting angiotensin II, suppressing the renin receptor, and thus protecting from arterial hypertension. The same effect is observed after acetate supplementation by producing a downregulation of local RAS in both heart and kidneys [41,42]. On the other hand, although microbial propionate suppresses angiotensin II, its intermediate succinate synthesis, under high glucose levels, triggers the kidney-specific G protein-coupled metabolic receptor, GPR91, that, in turn, leads to RAS over-activation. It is well known that the gut metabolite trimethylamine (TMA) and its oxide TMAO affect the cardiovascular system. Interestingly, it has been shown that, although TMAO infusion alone has no effect on blood pressure, when combined with even a low dose of angiotensin II, it prolongs the hypertensive effect [43]. Notably, in experimental models, chronic treatment with TMAO reduces AT1 receptor activity but increases AT2 expression [44,45]. Interestingly, Angiotensin II can alter the gut microbiota environment and metabolites (increase in Firmicutes/Bacteroidetes ratio) [46], which might affect the heart and other organs. On the contrary, there are reports suggesting that treatment with the ACE inhibitor modifies intestinal barrier permeability and consequently decreases TMA leaking into the circulation [47].

It is, however, notable that, since we do not know if the appropriate drug administration completely blocks the RAS system in patients with heart failure is questionable if the produced gut microbiota metabolites may prolong RAS activity with unknown consequences. Indeed, there are reports suggesting that gut microbiota metabolites can alter RAS function and gut sympathetic activity, promoting several cardiovascular diseases [48,49]. Thus, in keeping with what we know so far, the interaction between gut microbiota and RAS inhibitors is under investigation, and the information on this topic, although suggestive of a strong relationship, is still incomplete and speculative. Moreover, our knowledge regarding the effect of injectable drugs on the gut microbiota, and vice versa, is limited.

### 2.2. Sympathetic Activity

In patients with heart failure and reduced ejection fraction, the sympathetic activity is over-expressed and, therefore, the use of b-blockers, a medication that can reduce all-cause mortality, is necessary [50]. The sympathetic nervous system is found across the whole human body, including in the intestine. Indeed, this system innervates the stomach, small and large intestine, rectum, myenteric and submucosal plexi, and gastrointestinal blood vessels, and it controls intestinal function, secretion absorption, etc. [51,52]. Interestingly, the over-activity of the sympathetic nervous system promotes inflammation [53], another pillar of heart failure syndrome, facilitating the leakage of intestinal metabolites that, in turn, due to their harmful effects, [54], affect the heart. Importantly, the vagal innervation is located close by, sharing the same cells, and thus both arms of the autonomic nervous system are affected [55]. However, it is unknown which arm is majorly affected. It is of high interest to understand the newly discovered gut–autonomic nervous system activity [56,57,58], which represents the interference of the neurohormonal axis with the intestinal secretion, motility, immunity, and permeability [59,60,61,62]. Indeed, it has been suggested that metoprolol treatment for a long time may influence microbial composition and hence gastrointestinal tract dysbiosis, promoting arterial hypertension. The same is true for atenolol, which affects numerous metabolic processes in addition to the beta-adrenergic antagonism [63], whereas nebivolol bioavailability is low due to its limited intestinal permeability [64]. Bisoprolol, Nadolol, Pindolol, and Talinolol demonstrate slow distribution in a toxic environment, and it seems that this is the case when gut dysbiosis occurs [65].

In this context, the effects of beta-blockers on the gut microbiota–autonomic nervous system interference warrant further investigation.

### 2.3. Sodium-Glucose Cotransporter-2

Several studies have proved the beneficial effect of sodium-glucose cotransporter-2 (SGLT2) inhibitors, and thus both Dapagliflozin and Empagliflozin are strongly recommended for heart failure treatment [66,67]. The relation between the intestinal microbe’s changes, following the SGLT2 inhibitors’ administration, is well known [68,69]. In keeping with previous reports, it has been shown that the administration of Luseogliflozin, an SGLT2 inhibitor in mice, increases SCFA, which, as mentioned previously, has a beneficial effect on the heart [70]. The path that these drugs use, probably via gut-microbes, to increase SCFA production is not known, although it has been suggested that they may inhibit the absorption of simple sugars, along with increasing the group of microbes that are involved in the production of SCFA. At the same time, they have demonstrated a protective effect on the intestinal mucosa that reduce the inflammatory influx of different gut metabolites that are produced within the intestine, such as lipopolysaccharides and endotoxins, thus protecting the whole body from their toxic effects [70,71]. The therapeutic use of SGLT2 in heart failure has been part of the optimal medical treatment quite recently, and thus information regarding its action is under investigation and any conclusion in this respect might be speculative.

### 2.4. Diuretics

In patients with heart failure, a substantial number of patients demonstrate a deterioration of the already-existing kidney dysfunction. The cause of this kidney dysfunction seems to be the low output or/and high venous pressure leading to a low glomerular filtration rate. As the renal function deteriorates, higher doses of diuretics are needed. In patients with heart failure, the low glomerular filtration favors intestine metabolite gathering, alters the epithelial barrier of intestine promoting a vicious circle that increases gut permeability and the flow of toxic metabolites that in turn affect homeostatic process leading to an increase in mortality [72,73,74]. On the other hand, changes in gut microbiota lead to a change in uremic metabolites and an increase in uremic toxins that, in turn, deteriorate renal function and affect cardiac function [75,76]. In this respect it is reasonable to suppose that in some patients with heart failure although under diuretic therapy there might be still a congestion and the dose of diuretic therapy must be increased as tolerated. In the case of diuretic therapy, for instance, regarding the most used furosemide, we have to consider that the the gut microbes’ metabolites, along with the reduced absorption of proteins [77] that are necessary for furosemide action, are altered, and hence the diuretic action is diminished, requiring higher doses. This is in keeping with the recently published guidelines for heart failure treatment, where an increase of the diuretic dose, almost 2.5 times the chronic daily oral dose, has been suggested [50]. The use of diuretics is mostly essential for patients with heart failure presenting with congestion; however, little is known about their interaction with the gut microbiota and, therefore, no final conclusion can be reached.

In patients with heart failure, it must be taken into account that cardiorenal syndrome might affect both gut microbiota dysbiosis and renal function. Despite the recommended use of diuretics, it must be noted that the gut microbiota are connected to several organs, including the kidneys. The activation of immune cells, leading to a low-grade inflammatory reaction, affects the kidneys. Moreover, the peripheral nervous system alters neural inputs to the kidney, thus promoting further kidney dysfunction [78]. Additionally, it has been suggested that there is a specific intestinal flora that has a causal relationship with the incidence and progression of chronic kidney disease at the level of gene prediction [79].

## 3. Other Drugs Used in Heart Failure Treatment

A report using the metagenomics sequencing of stools in patients with heart failure found that several cardiac medicines, including glucosides, statins, and platelet aggregation inhibitors, can affect gut microbiota composition [80]. This has been confirmed by others, although the responsible mechanisms are not clearly reported [81]. The administration of digoxin can be converted by Eggerthella lenta into an inactive metabolite in a substantial number of patients (10%), thus limiting the amount of drug reaching the target tissue [82], whereas an arginine-rich diet demonstrates the drug–blood fluctuation of this specific drug [83], thus limiting its action. Aspirin is a drug used mainly in patients with coronary artery disease. There are reports suggesting its effect on the intestinal microbe’s composition, while, at the same time, bacterial communities can affect aspirin metabolism, altering its bioavailability and, therefore, its action [84,85]. The same is true for Amlodipine, a calcium channel blocker that reduces the amount of active drug reaching the tissues through pre-systemic metabolism dehydrogenation [86]. Concerning Statins, it seems that a common drug used in patients with heart diseases affects gut microbiota composition, has a close relation with intestinal microbe’s community demonstrating an inter-individual response variation, showing reduced quantity of active drug [30,80,87,88]. Indeed, it has been suggested that Statin users have compositionally differing microbiotas from non-users [89]. Notably, Statin therapy activates the inflammation process through gut leakage that leads to adverse effects, especially on the neuromuscular junction [90].

## 4. Suspected Mechanisms for Cardiac Drugs–Gut Microbiota Interplay

The interdependence of commonly used drugs on gut microbiome composition, and vice versa, has been well established [91]. Indeed, it has been shown that the microbiota, by leading to a chemical transformation of the gut microbiota environment, can change the efficacy of drugs [92]. In patients with heart failure, we also have to take into consideration that both comorbidities and polypharmacy play an important role in this interaction, altering the gut microbe’s composition, intestinal endocrine and paracrine function, mucosa absorption capability, and, ultimately, pharmacokinetics or pharmacodynamics and potential drug action on remote tissues. Several factors are involved in the metabolic process of a drug [93]. Notably, gut microbiota composition alteration is correlated with heat failure, with preserved ejection fraction related to comorbidities, fibrosis, and endothelial dysfunction [94,95,96]. However, although the principal role of endothelium in patients with heart failure has been demonstrated [97], there are reports questioning the role of gut microbiota modifications on arterial stiffness [98].

Gastrointestinal lumen status, including mucosa function and permeability, PH, grade of intestinal congestion, etc., can affect oral drug bioavailability in a different way, showing different actions depending on the gut microbe’s microenvironment [30]. Indeed, the intestine, with its multiple functions, is involved in the different steps of the metabolism and transportation of the drug from the intestinal lumen to the blood stream. By changing hydrophilic drugs to more hydrophobic compounds, the intestine ‘factory’ facilitates their absorption through the intestinal mucosa [93], including toxic metabolites [99]. This action is not uniform for each patient and depends on the inter-individual microbe’s difference [100], leading to a variation of drug responses [101]. In patients with heart failure, there is an imbalance in the intestinal microbial homeostatic environment, the so-called dysbiosis [4], that, in turn, increases the rate of cardiovascular diseases by producing toxic biochemical substances and causing the inactivation of oral drug action [99]. Gut microbiota consist of a variety of millions of microorganisms that produce a diversity of enzymes that metabolize most of the drugs with uncertain effects, accelerating or reducing their effect on the body’s function [28]. Notably, the composition of this new ‘organ’ is not uniform in all individuals and depends on several factors: host genetic variation, diet, lifestyle, xenobiotics, and medications [4]. This suggests that the effect of a drug on the intestinal microbe’s environment, and vice versa, is not uniform, nor is it stable. Therefore, the interplay between the oral drug and the gut microbiota may behave differently under the same circumstances, thus giving different results. In this respect, it is reasonable to suggest that a specific drug may act differently in individual patients with heart failure syndrome, thus explaining, to some extent, the remaining high mortality and morbidity rates, despite that patients are under he best medical treatment. For instance, it has been shown that the action of Diltiazem, after long-term administration, is increased [102], thus promoting a stress-like condition for microbes that, in turn, facilitates gene transfer from one species to another with unknown effects on cardiac drugs [103]. Based on this conception, it seems that any drug acts differently in each patient, showing a variability that has not been well understood until now. Thus, we have to try to understand better the drug–gut interplay and try to pass to the so called personalized and precision medicine after having known the exact metabolic effect of gut microbiome’s status. In this respect, a new era is emerging based on pharmaco-microbiomics and pharmacogenomics, both of which try to uncover gut microbiome traces and will probably help us to find the right therapeutic regimen. However, this is not an easy task to resolve because the gut microbiota are a dynamic ‘organ’, different in each patient, in a process of adaptation to changing cardiac and extracardiac conditions. In this respect, bacterial genome evaluation may improve our understanding of the gut microbiota [28].

Despite difficulties deciphering the mechanisms of the oral drug–gut microbiota interplay, the administration of certain drugs, and substances that can reduce the grade of intestinal dysbiosis, improves the microbe’s composition and restores the normal intestinal environment. In this respect, it has been shown that Helicobacter pylori improves the pharmacodynamics of Levodopa, facilitating its absorption [104]. Accordingly, it has been suggested that probiotics, fecal transplants, enzymes, diet modification, and exercise training can also modify the gut microbiome’s biochemical actions and, consequently, the effects of oral drugs [104,105]. Staying on the same path, a new concept is introduced, called quorum sensing, that is a process of cell–cell communication that acts through small molecules called auto-inducers, that might affect this drug–microbiota interaction [106,107,108,109]. However, we must note that we do not know if preservatives in oral medication affect intestinal flora, or if it is a compound. Indeed, if the bioavailability rate is poor, it passes through the intestinal tract without being absorbed. This is also true for patients with heart failure with preserved ejection fraction, in whom the exact mechanisms are not clear, suggesting that we have to separate patients into different phenotypes in order to find the optimal medical treatment [94].

### Cardiac Drugs–Gut Microbiota Interplay in Advanced, and in Different Types of, Heart Failure

Heart failure is a complex syndrome, and several factors must be taken into account in order to better understand the cardiac oral drugs–gut microbiota interplay. For example, this interplay in stable patients with recommended therapy could be different from that in patients with advanced or refractory heart failure, which is obviously due to the different stage of mucosa ischemia/congestion and to the imbalance of the intestinal microbial homeostatic environment, the so-called dysbiosis. Ischemia and/or congestion alter intestinal mucosa permeability, the intestinal barrier function is reduced, and the drug–microbiota interplay becomes abnormal. Additionally, several other organs (kidneys, endothelium, liver, etc.) are at play, as the syndrome progresses to a more severe situation. In this respect, right-side heart dysfunction could lead to the congestion of several organs including kidneys, body swelling, gut mucosa, etc., due to the increased venous pressure observed in this type of heart failure. Thus, a vicious circle arises consisting of the interplay among reduced right-heart function, renal dysfunction, endothelial dysfunction, altered gut mucosa barrier permeability, etc. This is an interplay that could alter the orally administrated cardiac drugs’ action, changing the drug bio-viability and absorption. Additionally, as the syndrome aggravates patients with heart failure may demonstrate abdominal discomfort, lack of appetite, alteration of several hormones including growth hormone, leptin, etc., leading to a negative imbalance between anabolic and catabolic metabolism and hence cachexia. A situation that alters the cardiac drugs–microbiota interplay [110].

Heart failure is a clinical syndrome that can be sub-divided, according to left ventricular ejection fraction, into three types—reduced (<40%), mildly reduced (41–49%), and preserved (>50%) left ventricular ejection fraction. Regardless of the phenotypic presentation, it seems that the basic cause for altered drug–gut microbiota interplay is the dysfunctional intestinal mucosa (due to ischemia and/or congestion) and the changes in the gut microbiota environment. However, it must be noted that heart failure is a syndrome in which other organs are also involved (kidneys, liver, endothelium, etc.), that can participate in the abnormal response of the drug administration. Indeed, intestinal microorganisms in collaboration with hepatic enzymes can convert fatty acids (e.g., choline, L-Carnitine) to trimethylamine N-oxide, a product that can promote several pathologies including platelet dysfunction, thrombotic events, and coronary artery disease. Moreover, in patients with heart failure, cardiorenal syndrome is very often present and might affect both gut microbiota dysbiosis, renal and hepatic dysfunction, and, in collaboration with gut dysbiosis, may alter drug actions. In any case, it seems that the interaction between drug action and gut microbiota is based principally on the severity and the duration of heart failure, regardless of the phenotypic presentation, since all types of heart failure cause congestion/ischemia and, hence, multi-organ dysfunction.

However, it must be noted that the pathophysiologic background underlying the different groups of patients could affect the microbiota. For example, neurohormonal modulation is able to improve the outcome of patients with LVEF <40% but not of those with greater LVEF. The improved outcome is associated with reverse remodeling, an improvement in both LVEF and hemodynamic status (i.e., congestion, filling pressure). This could influence the relationship between drugs and microbiota across the different values of LVEF.

In conclusion, it seems that the interplay between oral cardiac drug administration and the gut microbiota is a pragmatic concept (Figure 1). In patients with heart failure, the gut microbiota ecosystem is altered and this, in turn, can affect oral cardiac drug action. However, the actual mechanism of this interaction is not yet known. Further studies are required to determine whether interventions in the gut microbiota may potentiate the currently available drug effects and provide novel pathways for drug discovery.

## Figures and Tables

**Figure 1 ijms-25-01716-f001:**
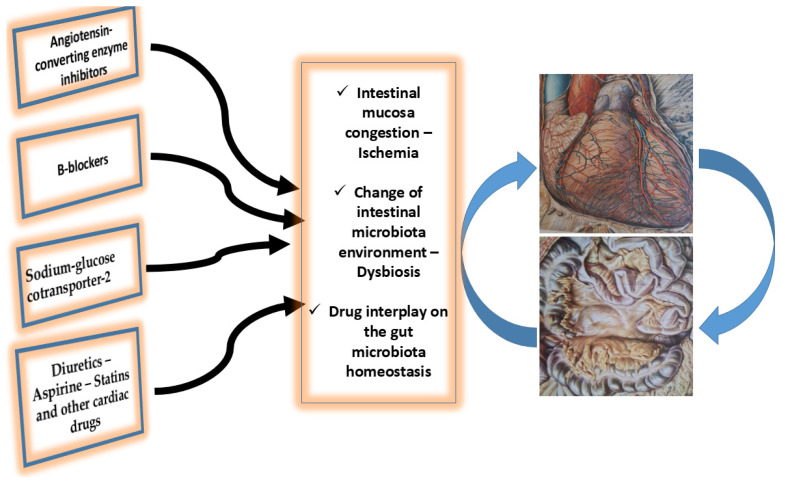
Interplay between the main oral cardiac drugs administration, heart, and gut microbiota in chronic heart failure patients. Intestinal mucosa congestion, its ischemic status promotes leakage of toxic substances. Additionally, the alteration of the intestinal microbes, and the dysbiosis and intestinal homeostasis alteration are the main factors of the drug–gut microbiota interplay. See text for more details.

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
