# Peer review of "Oral Cardiac Drug–Gut Microbiota Interaction in Chronic Heart Failure Patients: An Emerging Association"

_ijms, 2024, doi:10.3390/ijms25031716_

Round 1
Reviewer 1 Report
Comments and Suggestions for Authors
Reviewing the manuscript entitled, “Oral Cardiac Drug – Gut Microbiota Interaction in Heart Failure Patients: An Emerging Association” by Paraskevaidis I et al., this focuses on relationship between oral heart failure drugs and their effects on intestinal flora. The authors should consider the following concerns.
First of all, the authors should describe the basic mechanisms of oral drugs and their effects on the intestinal flora. Moreover, you should state why the focus is on chronic heart failure and its oral medications. Do preservatives in oral medications affect the intestinal flora? Or is it a compound? If the bioavailability rate is poor, it passes through the intestinal tract without being absorbed. Does this affect the intestinal flora?
As the authors state in the section on RAS inhibitors, if RAS inhibition itself has an effect, I believe that injectable drugs should also have an effect.
There are currently three categories of chronic heart failure: HFrEF, HFpEF, and HFmrEF. Although the authors called heart failure, only HFrEF is mentioned.
The content is almost narrative and lacks scientific basis.
Comments on the Quality of English LanguageMinor editing of English language required
Author Response
We do wish to thank this reviewer for his fruitful commends
Suggestion: First of all, the authors should describe the basic mechanisms of oral drugs and their effects on the intestinal flora.
Response: At the ‘Suspected mechanisms for cardiac drugs-gut microbiota interplay’, section 3, it is written: ….. ‘Indeed it has been shown that the microbiota by leading to a chemical transformation of the gut microbiota environment can change the efficacy of drugs.90
Moreover at the same section is written: ‘Notably, the composition of this new ‘organ’ is not uniform in all individuals depending on several factors; host genetic variation, diet, lifestyle, xenobiotics, and medications 4, suggesting that the effect of a drug on intestinal microbe’s environment and vice-versa is not uniform neither is stable and therefore the interplay oral drug-gut microbiota may behave differently under the same circumstances giving thus different results. In this respect, it is reasonable to suggest that a specific drug may act differently in individual patients with heart failure syndrome ……….
Moreover at the same section it is also written: ‘…..it seems that any drug acts differently in every each patient showing a variability that it is not well understood till now. Thus, we have to try to understand better the drug-gut interplay and try to pass to the so called personalized and precision medicine after have known the exact metabolic effect of gut microbiome’s status’
Suggestion: Moreover, you should state why the focus is on chronic heart failure and its oral medications.
Response: Following his suggestion we add the term chronic at the title, abstract etc. Additionally, at the end of the introduction we state: ‘Moreover, we will focus on oral drug administration and not on the injectable drugs since the former are the main therapeutic approach in every day clinical practice in patients with chronic heart failure’.
Suggestion: Do preservatives in oral medications affect the intestinal flora? Or is it a compound? If the bioavailability rate is poor, it passes through the intestinal tract without being absorbed. Does this affect the intestinal flora?
Response: We fully agree with this suggestion – question and therefore at the ‘Suspected mechanisms for cardiac drugs-gut microbiota interplay ‘ section 3 it is written: ‘However, we have to notice that we don’t know if preservatives in oral medication affect intestinal flora or if it is a compound. Indeed, if the bioavailability rate is poor it passes through the intestinal tract without being absorbed’
Suggestion: As the authors state in the section on RAS inhibitors, if RAS inhibition itself has an effect, I believe that injectable drugs should also have an effect.
Response: We thank this reviewer for this suggestion. We do believe the same however there are no strong data to support this. Thus at the end of the introduction it is written: Moreover, we will focus on oral drug administration and not on the injectable drugs since the former are the main therapeutic approach in every day clinical practice in patients with chronic heart failure.
Moreover at the end of 1a section it is written: …..’ our knowledge regarding the effect of injectable drugs on gut microbiota and vice versa is limited’.
Suggestion There are currently three categories of chronic heart failure: HFrEF, HFpEF, and HFmrEF. Although the authors called heart failure, only HFrEF is mentioned.
Response: At the section ‘oral drugs’ it is written ‘For patients with mildly reduced or preserved ejection fraction, quadruple therapy could be utilized but not strongly recommended’ 34
Moreover at the section 3 ‘Suspected mechanisms for cardiac drugs-gut microbiota interplay’ it is written: ‘Notably, gut microbiota composition-alteration is correlated with heart failure with preserved ejection fraction being related to comorbidities, fibrosis and endothelial dysfunction. However, although the principal role of endothelium in patients with heart failure preserved ejection fraction has been demonstrated there are reports questioning the role of gut microbiota modifications on arterial stiffness’. 92-96
Suggestion: The content is almost narrative and lacks scientific basis.
Response: We are deeply sorry for not have convinced this reviewer and forced him to express his opinion in such way.
Reviewer 2 Report
Comments and Suggestions for Authors
The submitted review is of great interest and overall well organised. However, it lacks few important concepts. My comments are mostly minor:
- Figure 1 is oversized and oversimplified, I would redesign the figure reducing the scale of images and increasing that of text. Furthermore, I would add more interactions (see below).
- Paragraph 1. b. Sympathetic activity Paragraph should be modified according to other paragraphs of the section. Please use the drug class name to be consistent.
- Although medications play a crucial role and is the main topic of the review, few confounding factors should be taken into account. For example the concomitant presence of kidney dysfunction (due to cardio renal syndrome) and its impact in modifying gut microbiota (see PMID: 36678231). And the degree of vascular dysfunction in heart failure and gut microbiota (see PMID: 35743626, PMID: 11868041).
- Finally, patients with heart failure very commonly have dyslipidemia or an history of ischemic cardiac disease. In both cases, they are treated with statins and/or ezetimibe. I would definitely explore also this class of drugs.
Author Response
We do wish to thank this reviewer very much for his valuable suggestions
Suggestion: Figure 1 is oversized and oversimplified, I would redesign the figure reducing the scale of images and increasing that of text. Furthermore, I would add more interactions (see below).
Response: We change Figure 1, by reducing the scale and give more information on the gut – drug interplay and have increase the text accordingly.
Suggestion. Paragraph 1. b. Sympathetic activity Paragraph should be modified according to other paragraphs of the section. Please use the drug class name to be consistent.
Response: At the 1.b. section ‘Sympathetic activity’, it is written: ‘Indeed, it has been suggested that metoprolol treatment for long time may influence microbial composition and hence gastrointestinal tract dysbiosis, promoting arterial hypertension. The same is true for atenolol that affects numerous metabolic processes in addition to the beta-adrenergic antagonism 61 whereas nebivolol bioavailability is low due to its limited intestinal permeability. 62 Bisoprolol, nadolol, pindolol and talinolol demonstrate slow distribution in a toxic environment and it seems that this is the case when gut dysbiosis occurs. 63
Suggestion: Although medications play a crucial role and is the main topic of the review, few confounding factors should be taken into account. For example the concomitant presence of kidney dysfunction (due to cardio renal syndrome) and its impact in modifying gut microbiota (see PMID: 36678231). And the degree of vascular dysfunction in heart failure and gut microbiota (see PMID: 35743626, PMID: 11868041).
Response: At 1.d. ‘Diuretic’ section it is written. ‘In patients with heart failure it must be taken into account the cardio renal syndrome that might affect both gut microbiota environment and renal function. Despite the recommended use of diuretics it has to be notice that gut microbiota is in connection with several organs including kidneys. Through the activation of immune cells leading to a low-grade inflammatory reaction affect the kidneys. Moreover, peripheral nervous system alter neural inputs to the kidney, promoting thus a further kidney dysfunction.76 Additionally, it has been suggested a specific intestinal flora that has a causal relationship with the incidence and progression of chronic kidney disease at the level of gene prediction. 77
Moreover, at section 3 ‘Suspected mechanisms for cardiac drugs-gut microbiota interplay’ it is written ‘Notably, gut microbiota composition-alteration is correlated with heat failure with preserved ejection fraction being related to comorbidities, fibrosis and endothelial dysfunction. 92-94 However, although the principal role of endothelium in patients with heart failure has been demonstrated 95 there are reports questioning the role of gut microbiota modifications on arterial stiffness’. 96
Obviously the suggested reports PMID: 36678231, PMID: 35743626, PMID: 11868041 are incorporated.
Suggestion: Finally, patients with heart failure very commonly have dyslipidemia or an history of ischemic cardiac disease. In both cases, they are treated with statins and/or ezetimibe. I would definitely explore also this class of drugs
Response: At the section 2, ‘Other drugs used in heart failure treatment’ it is written. ‘Concerning statins a common drug used in patients with heart diseases it seems that affect gut microbiota composition, have a close relation with intestinal microbe’s community demonstrating an inter-individual response variation, showing reduced quantity of active drug. 28,78,85,86 Indeed it has been suggested that Statin users have compositionally differing microbiotas from nonusers’. 87 Notably, statin therapy activates the inflammation process through gut leakage that lead to adverse affects specially on the neuromuscular junction’. 88
Round 2
Reviewer 1 Report
Comments and Suggestions for Authors
The content is still narrative and lacks scientific basis.
The authors mentioned “For patients with mildly reduced or preserved ejection fraction, quadruple therapy could be utilized but not strongly recommended.”
This statement is difficult to understand. Even with HFpEF, the patients take oral heart failure medications such as SGLT2, diuretics. Do preservatives in oral medications affect the intestinal flora? Or is it a compound? If the bioavailability rate is poor, it passes through the intestinal tract without being absorbed. Does this affect the intestinal flora? The authors do not adequately address this concern.
If oral heart failure drugs have a significant impact on the intestinal bacterial layer, what are the specific strategies for treating heart failure patients?
Author Response
The content is still narrative and lacks scientific basis.
The authors mentioned “For patients with mildly reduced or preserved ejection fraction, quadruple therapy could be utilized but not strongly recommended.”
This statement is difficult to understand. Even with HFpEF, the patients take oral heart failure medications such as SGLT2, diuretics. Do preservatives in oral medications affect the intestinal flora? Or is it a compound? If the bioavailability rate is poor, it passes through the intestinal tract without being absorbed. Does this affect the intestinal flora? The authors do not adequately address this concern.
RESPONSE: To our knowledge there are limited data concerning these questions. Accordingly at the ‘Suspected mechanisms for cardiac drugs-gut microbiota interplay ‘ section 3 it is written ‘However, we have to notice that we don’t know if preservatives in oral medication affect intestinal flora or if it is a compound. Indeed, if the bioavailability rate is poor it passes through the intestinal tract without being absorbed’
At the same section it is also written: This is also true for patients with Heart Failure preserved Ejection Fraction in whom the exact mechanisms are not clear, suggesting that we have to separate patients into different phenotypes in order to find the optimal medical treatment. REF 92: Yu, W.; Jiang, Y.; Xu, H.; Zhou, Y. The Interaction of Gut Microbiota and Heart Failure with Preserved Ejection Fraction: From Mechanism to Potential Therapies. Biomedicines 2023, 11, 442
If oral heart failure drugs have a significant impact on the intestinal bacterial layer, what are the specific strategies for treating heart failure patients?
RESPONSE: At the section 3 ‘Suspected mechanisms for cardiac drugs-gut microbiota interplay ‘ it is written: Accordingly, it has been suggested that probiotics, fecal transplant, enzymes, diet modification, exercise training can also modify gut microbiome its biochemical actions and consequently oral drug effect.102,103 Staying on the same path, a new concept is introduced; quorum sensing, that is a process of cell–cell communication that acts through small molecules called auto-inducers that might affect this drug-microbiota interaction.104-107

Reviewer 2 Report
Comments and Suggestions for Authors
The paper has significantly improved and it is now suitable for publication. I endorse the publication of the manuscript.
Author Response
The paper has significantly improved and it is now suitable for publication. I endorse the publication of the manuscript.
RESPONSE: We thank very much this reviewer. The improvement is due to his suggestions

Round 3
Reviewer 1 Report
Comments and Suggestions for Authors
In V3 revised version, the sentence before the conclusion has been changed, but there is no change in the content. Overall, it lacks scientific basis and is narrative. It is also a problem that HFpEF is not considered in the definition of chronic heart failure, and even in HFpEF patients have no choice but to take Beta blockade or SGLT2 inhibition, and I still feel that the author's description is strange.
Author Response
Response: there is no definition in chronic heart failure that incorporate HF r,m,p EF. This is a phenotypic division. Regarding the therapeutic approach for HFpEF the recommendation for SGLT2 is IIa level of evidence B and not I and level of evidence A.
This reviewer has to understand what he reads. In fact his behavior is really strange and scientifically incorrect